# Variability in the Clinical Course of COVID-19 in a Retrospective Analysis of a Large Real-World Database

**DOI:** 10.3390/v15010149

**Published:** 2023-01-03

**Authors:** Robert Flisiak, Piotr Rzymski, Dorota Zarębska-Michaluk, Przemysław Ciechanowski, Krystyna Dobrowolska, Magdalena Rogalska, Jerzy Jaroszewicz, Anna Szymanek-Pasternak, Marta Rorat, Dorota Kozielewicz, Justyna Kowalska, Ewa Dutkiewicz, Katarzyna Sikorska, Anna Moniuszko-Malinowska

**Affiliations:** 1Department of Infectious Diseases and Hepatology, Medical University of Białystok, 15-540 Białystok, Poland; 2Department of Environmental Medicine, Poznan University of Medical Sciences, 60-806 Poznań, Poland; 3Department of Infectious Diseases, Jan Kochanowski University, 25-317 Kielce, Poland; 4Department of Paediatrics and Infectious Diseases, Regional Hospital in Szczecin, 71-455 Szczecin, Poland; 5Collegium Medicum, Jan Kochanowski University, 25-317 Kielce, Poland; 6Department of Infectious Diseases and Hepatology, Medical University of Silesia in Katowice, 41-902 Bytom, Poland; 7Department of Infectious Diseases and Hepatology, Wrocław Medical University, 51-149 Wrocław, Poland; 8Department of Forensic Medicine, Wrocław Medical University, 50-367 Wroclaw, Poland; 9Department of Infectious Diseases and Hepatology, Faculty of Medicine, Collegium Medicum in Bydgoszcz, Nicolaus Copernicus University, 87-100 Torun, Poland; 10Department of Adult’s Infectious Diseases, Medical University of Warsaw, Hospital for Infectious Diseases, 02-091 Warsaw, Poland; 11Division of Tropical and Parasitic Diseases, Faculty of Health Sciences, Medical University of Gdańsk, 80-210 Gdańsk, Poland; 12Department of Infectious Diseases and Neuroinfections, Medical University of Białystok, 15-809 Bialystok, Poland

**Keywords:** COVID-19, SARS-CoV-2, Omicron, clinical picture

## Abstract

The COVID-19 pandemic proceeds in waves, with variable characteristics of the clinical picture resulting from the evolution of the SARS-CoV-2 virus. This study aimed to compare the epidemiological characteristics, symptomatology, and outcomes of the disease in patients hospitalized for COVID-19 during periods of different variants dominance. Comparing the periods of dominance of variants preceding the Delta variant, the Delta period was characterized by a higher share of hospitalized females, less frequent comorbidities among patients, and a different age distribution. The lowest need for oxygen therapy and mechanical ventilation was observed under Omicron dominance. The triad of classic COVID-19 symptoms, cough, fever, dyspnoea, and fatigue, were most prevalent during the Delta period, and significantly less common under the Omicron dominance. During the Omicron period, nearly twice as many patients as in the previous periods could be discharged from the hospital within 7 days; the overall 28-day mortality was significantly lower compared to that of the Delta period. It also did not differ between periods that were dominated by the BA.1 and BA.2 subvariants. The study indicates that the Omicron SARS-CoV-2 variant that dominated between January and June 2022 caused a disease which resembled the common cold, and was caused by seasonal alpha and beta-coronaviruses with a low pathogenicity for humans. However, one should note that this effect may not only have been related to biological features of the Omicron lineage, but may additionally have been driven by the increased levels of immunization through natural infections and vaccinations, for which we could not account for due to a lack of sufficient data.

## 1. Introduction

At the end of December 2019, the first cases of the illness caused by the severe acute respiratory syndrome coronavirus 2 (SARS-CoV-2) were reported in the Chinese province of Wuhan. Very quickly, the epidemic of SARS-CoV-2 infections ceased to be local and spread worldwide, resulting in the World Health Organization (WHO) declaring a COVID-19 pandemic on 11 March 2020 [1].

The number of infections grew exponentially; by the end of September 2022, more than 614 million had been reported worldwide, and more than 6.5 million people had died from COVID-19 [2]. Initially, infections worldwide were caused by a wild type of the virus, but then SARS-CoV-2 changed over time, creating new strains. The WHO has been monitoring and evaluating the evolution of SARS-CoV-2 since January 2020. It prompted the definition of specific variants of interest (VoI) for those associated with an increased burden on the global public, and variants of concern (VoC) responsible for subsequent waves of pandemics. Currently, five VoC have been identified and named with letters of the Greek alphabet, Alpha (B.1.1.7), Beta (B.1.351), Gamma (P.1), Delta (B.1.617), and Omicron (B.1.1.529) [3,4]. 

Available observations indicate that waves of COVID-19 caused by successive SARS-CoV-2 VoC differed in transmissibility, the severity of the clinical course of the disease, and mortality rates [5,6,7,8]. Infection with subsequent VoC were associated with the reduced effectiveness of antibodies generated due to the infections with previous variants, and decreased the activity of some monoclonal antibodies [9,10,11]. The reduced efficacy of vaccines prepared based on studies with the SARS-CoV-2 wild-type strain became the basis for developing formulations with an updated composition [12]. The most pronounced differences were noted for the last two variants, Delta and Omicron. The Delta variant was first detected in India in October 2020, and had over 25 mutations compared to the wild-type SARS-CoV-2 strain, 9 of which involved the S protein (plus D614G), including the hallmark L452R, P681R, and T478K substitutions. However, it was quickly replaced by a subsequent VoC, Omicron, first identified in November 2021, and became globally dominant as early as December 2021 [13]. Several Omicron subvariants soon followed, such as 21K (BA.1) and 21L (BA.2), now replaced by subvariants 22A (BA.4) and 22B (BA.5) with even greater transmissibility [14].

The present real-world analysis aimed to compare the epidemiological characteristics, baseline clinical symptoms and laboratory parameters, the clinical course of SARS-CoV-2 infection, and the outcomes of the disease in patients hospitalized for COVID-19 in Poland during the periods of pre-Delta, Delta, and Omicron variant dominance. These periods were selected, as previous studies have shown that Delta and Omicron variants had the greatest impact on the clinical image of SARS-CoV-2 infections. However, the majority of the research focused on binary comparisons of particular variants [15]; meanwhile, longitudinal investigations encompassing larger sample sizes and more than two periods dominated by different SARS-CoV-2 lineages are scarce, and were often limited to specific patient groups, e.g., the pediatric population [16]. The present study is the first to demonstrate clinical changes in infection severity among patients hospitalized in various medical units across Poland.

## 2. Materials and Methods

### 2.1. Data Collection

The data for this study were collected retrospectively, using the observational, nationwide SARSTer database supported by the Polish Association of Epidemiologists and Infectiologists. The database includes 44 Polish centers that collected data on the clinical characteristics and treatment of 11,898 patients hospitalized due to COVID-19 since the beginning of the pandemic. They were diagnosed and treated according to the applicable national recommendations for managing COVID-19 [17,18,19]. The data were analyzed in three pandemic periods 1 March 2020–30 June 2021 (defined in the paper as a pre-Delta period), 1 July 2021–31 December 2021 (defined as a Delta period), and 1 January 2022–30 June 2022 (defined as an Omicron period), which correspond to periods of dominance of variants preceding the appearance of the Delta variant, the dominance of the Delta variant and the Omicron variant, respectively. During these periods, the following number of infections were noted in Poland: 2,879,912, 1,228,205, and 1,906,777, respectively. These three periods were established based on sequences submitted by Polish laboratories according to the Global Initiative on Sharing All Influenza Data (GISAID), the most reliable database on SARS-CoV-2 variants prevalence in different regions of the world [20]. According to data available for our country, the defined pre-Delta period (1 March 2020–30 June 2021) was dominated by infections with Nextstrain clades 20A, 20B, and 20C (till February 2021), and the alpha variant (from February 2021). These variants did not reveal major differences in clinical outcomes [7,21]. During the Omicron period, two subvariants prevailed: BA.1 (January–February 2022) and BA.2 (March–June 2022). Patient characteristics included in each analyzed period were sex, age, and comorbidities. The course of the disease included the length of hospitalization, the need for oxygen therapy and mechanical ventilation, the clinical condition on admission based on oxygen saturation (SpO_2_), and the frequency of the most common symptoms. Information on vaccination status and history of previous infections with SARS-CoV-2 was unavailable in the database.

Additionally, the clinical course of the disease was assessed on admission to the hospital, and then after 7, 14, 21, and 28 days using an ordinal scale based on WHO recommendations, it was modified to the 8-point version to match the specificity of the Polish healthcare system and used in previous SARSTer research [11,22,23]. The score was defined as follows: (1) not hospitalized, no activity restrictions; (2) not hospitalized, no activity restrictions and/or not requiring oxygen supplementation at home; (3) hospitalized, and not requiring oxygen supplementation and not requiring medical care; (4) hospitalized, not requiring oxygen supplementation, but requiring medical care; (5) hospitalized, requiring normal oxygen supplementation; (6) hospitalized, requiring non-invasive ventilation with high-flow oxygen equipment; (7) hospitalized, for invasive mechanical ventilation or extracorporeal membrane oxygenation; (8) death. Mortality was assessed within 28 days of hospitalization by age, and in the subpopulations at the highest risk of death by age, baseline SpO_2_ < 91%, and comorbidities.

### 2.2. Statistical Analyses

Demographic and clinical data of patients were analyzed with Statistica v13.3 (StatSoft Inc., Tulsa, OK, USA). The frequencies of particular characteristics or events were compared between the pre-Delta and Delta periods, and the Delta and Omicron periods, using Pearson’s chi-square (χ^2^) test. The interval measures were compared using Student’s *t*-test. When *p* < 0.05, differences were deemed statistically significant.

## 3. Results

### 3.1. Demographic Characteristics of Patients Hospitalized during Different Periods of the Pandemic

The monthly average number of patients registered at hospitals was similar in pre-Delta and Delta periods (484 and 488, respectively) but dropped over two-fold to 205 under Omicron dominance. Significant differences existed between pandemic periods dominated by subsequent SARS-CoV-2 variants (Table 1). During the Omicron period, the age of the patients was significantly lower than that observed in the Delta period and the preceding one. This was mainly due to twice as frequent hospitalization of people under 20, although patients over 80 constituted a significantly higher percentage during the Omicron period compared to the preceding periods. Compared to the pre-Delta period, the Delta period was characterized by a higher share of hospitalized females, less frequent comorbidities among patients, and a different age distribution, with higher percentages of individuals <40 and >80 years, but lower of those 40–80 years. The age of deceased patients was higher during the Delta period compared to the pre-Delta period.

### 3.2. Clinical Features of Patients Hospitalized during Different Periods of the Pandemic

The characteristics of the clinical course of the disease in patients hospitalized during different periods revealed distinctive differences (Table 2). The greatest needs for oxygen therapy and mechanical ventilation were observed under Delta dominance, and the lowest needs occurred during Omicron dominance. During the Delta period, the share of patients with SpO_2_ < 91% at admission or presenting ARDS was greater than those in the pre-Delta and Omicron periods. Moreover, the analyzed periods differed in the frequency of the most common symptoms of COVID-19. The classic COVID-19 symptoms, cough, fever, dyspnoea, and fatigue, were most prevalent during the Delta period, and significantly less common under Omicron dominance. In turn, the latter was characterized by the highest share of patients with vomiting. Loss of smell and taste was most common during the pre-Delta period, and only sporadically present during the Omicron period.

As shown in Figure 1, during the Omicron period, as much as 36.5%, i.e., nearly twice as many patients as in the previous periods (18.3% and 19.5%), were discharged from the hospital within 7 days. 

After two, three, and four weeks, the percentages of patients discharged during the Omicron period was the highest (71.9%, 82%, and 85.3%, respectively), and the difference was statistically significant compared to the Delta period (Table 3). 

### 3.3. Mortality of Patients Hospitalized during Different Periods of the Pandemic

The overall 28-day mortality was the lowest during the Omicron period, and the difference was statistically significant compared to the Delta period. The same tendency was observed after dividing patients by age, but statistical significance only applied to patients aged 60–80 and over 80 years (Table 4). As shown, the mortality assessed in the subpopulations most at risk of death was significantly lower during the Omicron period than under the dominance of the Delta variant, only in patients over 80 years of age with a baseline SpO_2_ below 91% and hypertension (Table 4). On the other hand, in the pre-Delta period, the significantly lower mortality compared to the Delta period concerned patients with a baseline SpO_2_ below 91%, as well as individuals aged over 80 who, in addition to having an SpO_2_ < 91%, were diabetic.

Periods dominated by the BA.1 and BA.2 subvariants of Omicron lineage did not differ in frequency of patients with low oxygen saturation, or requiring oxygen therapy and mechanical ventilation. The 28-day mortalities were also similar during these two periods. The only difference was the older age, higher rate of patients with comorbidities, the occurrence of dyspnoea, and the frequency of individuals discharged within 7 days from admission during the domination of BA.2 (Table 5). 

## 4. Discussion

### 4.1. Clinical Course of Disease during Periods Dominated by Different SARS-CoV-2 Variants

The results of the present study indicate that during the dominance of the Omicron SARS-CoV-2 variant, significantly fewer patients were hospitalized. We also observed a tendency for a reduced absolute number of hospitalized patients during the Omicron wave when compared to the Delta vs. pre-Delta periods. A similar decrease in severity during the Omicron wave was observed in other studies [24,25,26,27,28]. There are two plausible explanations behind the decreased severity seen during the Omicron-dominated period that likely overlap with each other: (i) increased levels of immunization in the population due to vaccination and previous infections; and (ii) biological features of Omicron that translated into lower severity of infections, particularly decreased fusogenicity and preference of the endocytic route of cellular infection [29].

In the present study, the lower mean age of patients admitted to the hospital during the Omicron wave compared to previously analyzed periods resulted from a significantly higher percentage of those under the age of 20. The phenomenon of an unprecedented increase in the number of hospitalizations of children and adolescents in the Omicron surge has been pointed out by other authors [25,30,31]. In contrast, analyses that were conducted only in the adult population, and evaluated patients hospitalized in different waves of the pandemic in different countries including Poland, reported an increase in the mean age of patients during the Omicron wave [8,32]. The increased hospitalization of younger individuals during the Omicron period may have potentially resulted from increased transmissibility of this variant and lower vaccination rates in individuals aged <20 compared to other age groups [33].

The period of Omicron dominance compared to the pre-Delta and Delta periods was also characterized by a significantly lower rate of hospitalizations for patients aged 20–80; this difference was particularly pronounced in the 40–60 age group. However, during the Omicron wave, patients over 80 years of age were hospitalized significantly more often than in previous periods of the COVID-19 pandemic in Poland. The share of this population of patients also increased significantly during the Delta wave compared to the earlier analyzed pandemic season in Poland. The findings that elderly persons were hospitalized more often in the Omicron wave than in previous pandemic periods are consistent with published data from other countries [28,34]. It cannot be ruled out that in our analysis, some patients, especially the elderly, were admitted to the hospital primarily for conditions other than COVID-19, and SARS-CoV-2 infection was an associated circumstance, as other researchers have pointed out [34].

We documented that on admission to the hospital, patients in the Omicron wave showed a significantly better clinical condition compared to the previous pandemic periods; meanwhile, the highest percentage of those presenting the most severe clinical condition was recorded for the Delta wave, which supported other reports [6,25,28,34,35]. Omicron-infected patients significantly less frequently demonstrated cough, fever, shortness of breath, fatigue, headache, and only occasionally loss of smell and taste, i.e., symptoms considered at the beginning of the pandemic to be typical for COVID-19. The incidence of gastrointestinal symptoms such as nausea and diarrhea were comparable in the Omicron and Delta waves, while vomiting was reported significantly more often. The changes in the clinical picture of the disease reported in our study during the emergence and dominance of the Omicron strain were described in many other analyses that compared the severity of SARS-CoV-2 infections during periods dominated by different viral variants [25,30,36]. 

Furthermore, the results of our study regarding the milder course of the disease during the Omicron-dominant period, as manifested by a significantly lower percentage of patients requiring oxygen therapy and mechanical ventilation compared to both of the previous periods analyzed, are consistent with reports from other researchers [15,24,28]. It should be noted that, according to the results of our analysis, the highest rate of patients requiring oxygen supplementation and invasive mechanical ventilation was documented during the period of dominance of the Delta variant, which is consistent with observations from other studies [6,28,32,35]. The fact that during the dominance of BA.2 compared to BA.1, patients were older, more likely to have chronic diseases, but nevertheless were discharged from the hospital sooner and died less, generally indicates that BA.2 infections were milder.

According to the current analysis, the mortality rate decreased among patients hospitalized during the dominance of the Omicron variant in Poland, which was specifically noticeable among the elderly, despite the higher percentage of people with comorbidities during this pandemic period. In particular, a lower association of mortality with diabetes and cardiovascular diseases was found during the Omicron-dominated period. We reported the highest mortality among patients hospitalized during the period when the Delta variant dominated. This conclusion is not surprising, as available analyses indicate this thus far. The Delta variant infection was associated with the highest mortality, and our study confirms that this was also true for the Polish population [24,25,26,32,37]. Similarly, observations of reduced mortality during Omicron dominance are common [15,28,34]. It should be kept in mind that some data from other countries are difficult to extrapolate to Poland, due to differences in healthcare resource utilization, socio-behavioral conditions, and patient characteristics, including immunization status.

### 4.2. Potential Future Changes in Clinical Severity of SARS-CoV-2 Infections

Predicting the future clinical relevance of SARS-CoV-2 is challenging, although some plausible scenarios can be put forward. Firstly, it is not certain that future viral evolution will necessarily lead to a decrease in clinical severity. The Omicron variant does not cause higher viral loads in the respiratory tract than the Delta variant (with some research suggesting that these loads may even be lower) [38,39,40]; moreover, it has no greater affinity to the angiotensin-converting enzyme-2 receptor, and is less fusogenic [41,42]. Considering that SARS-CoV-2 is most transmissible prior to symptoms onset and at the beginning of the symptomatic phase [43], the mutation-enhancing viral loads could lead to superior transmission, yet be accompanied by more severe infections due to increased risk of hyperinflammation. However, in another scenario, SARS-CoV-2 may continue to evolve into a greater escape from infection- and vaccination-acquired (including variant-adapted vaccines) immunity. This could lead to its high transmissibility without a significant increase in severity, particularly if immune escape would mostly concern humoral and not cellular responses. It cannot be excluded that mutation-acquired novel routes of cellular infection may eventually affect the clinical severity of SARS-CoV-2. As shown, the Omicron variant can use cell-surface fusion and endosomal fusion, with the latter being a preferred route [44,45]. This considerably increases the number of cell types this variant could potentially infect. Although it did not translate into greater clinical severity for Omicron, it may become more relevant under the emergence of other mutations in the future. It is also plausible that SARS-CoV-2 will mutate in non-human hosts, and return to the human population via contact with farmed (e.g., mink) [46,47,48] or wild (e.g., white-tailed deer) animals [49,50,51]. As it has been shown, Omicron can use ACE2 receptors from a broader range of host species than other variants, including domestic poultry and mice [44]. The consequences of this process may or may not affect clinical severity, since mutation-driven adaptations to a new host may lead to decreased adaptation to the human environment [52,53].

### 4.3. Study Limitations and Strengths

The disregard of immunization status in this analysis is one of its limitations, of which we are aware. This was a factor whose impact on the number of patients with COVID-19 course requiring hospital treatment in the various waves of the pandemic was analyzed in many studies, with particular attention paid to the degree of vaccination [32,54]. However, immunization status is also a cumulative effect of previous SARS-CoV-2 infections, which would also need to be considered, and encompasses data from medical history and serological markers. In addition, for COVID-19 vaccinations, the protective effect depends on the number of doses received, and the time elapsed between the last dose and the SARS-CoV-2 infection. We assumed that with so many confounding factors, comparing patients simply by dividing them into vaccinated and unvaccinated would be unreliable. We also did not perform viral sequencing for the hospitalized patients, but instead separated different pandemic periods based on reliable genomic sequence surveillance data deposited in the GISAID. Moreover, rates of death due to COVID-19 may, to some extent, be influenced by additional variables, such as changes in healthcare system capacity or environmental factors (e.g., air quality) [55,56]. Since our analysis involved hospitalized patients, and thus the vast majority of symptomatic individuals, its results cannot be translated to the entire infected population in subsequent waves of the pandemic. We also did not specify the group of hospitalized patients in whom a COVID-19 diagnosis may have been accidental, and contributed to rather than caused hospitalization or death, a phenomenon observed especially during the dominance of the Omicron wave. Finally, retrospective data collection based on medical records may have caused potential bias. 

As a major strength of our study, we highlighted the analysis of data from a large real-world population from many different centers in our country, including pediatric facilities, which ensures nationwide coverage that increases generalizability. The centers included in the study managed patients based on the same national recommendations. For each patient, detailed information on the baseline characteristics and outcomes was available. What is important, in every case, we collected data for a period of 28 days from admission to the hospital, including follow-up monitoring after discharge from the hospital, unless there was prior death.

## 5. Conclusions

In summary, the study indicates that the Omicron SARS-CoV-2 variant dominating between January and June 2022 caused the disease with a course that is similar to the common cold disease caused by previously discovered coronaviruses, with low pathogenicity for humans. This is likely a result of two overlapping causes: (i) biological features of the Omicron variant that lead to lower disease severity and (ii) increased immunization levels in the population due to infections with previous variants and vaccinations. Nevertheless, the differences in the clinical picture compared to classic COVID-19 justify the recognition of the disease caused by the Omicron variant to be a separate disease entity, which could possibly be called COVID-22. However, one should bear in mind that the epidemiological situation is still dynamic, and it is challenging to predict whether further evolution of SARS-CoV-2 will be associated with decreased severity of the infection.

## Figures and Tables

**Figure 1 viruses-15-00149-f001:**
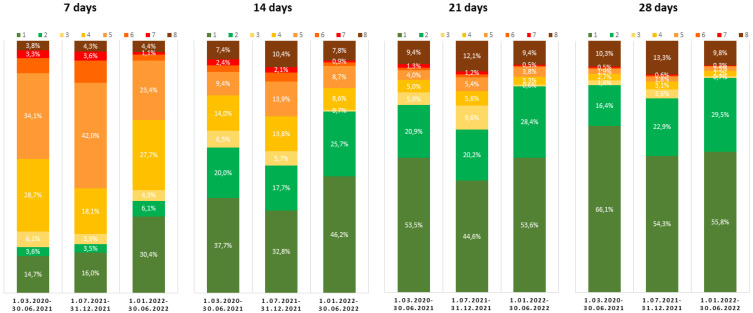
Ordinal scales after 7, 14, 21, and 28 days of hospitalization due to COVID-19 during periods with the dominance of particular SARS-CoV-2 variants. The scores were defined as follows: (1) not hospitalized, no activity restrictions; (2) not hospitalized, no activity restrictions and/or not requiring oxygen supplementation at home; (3) hospitalized, not requiring oxygen supplementation and not requiring medical care; (4) hospitalized, requiring no oxygen supplementation, but requiring medical care; (5) hospitalized, requiring normal oxygen supplementation; (6) hospitalized, requiring non-invasive ventilation with high-flow oxygen equipment; (7) hospitalized, for invasive mechanical ventilation or extracorporeal membrane oxygenation; (8) death.

**Table 1 viruses-15-00149-t001:** Characteristics of patients hospitalized during periods with the dominance of particular SARS-CoV-2 variants.

	Pre-Delta 1 March 2020–30 June 2021 *N* = 7742	Delta 1 July 2021–31 December 2021 *N* = 2929	*p* Delta vs. Pre-Delta	Omicron 1 January 2022–30 June 2022 *N* = 1227	*p* Omicron vs. Delta
Females/males, *n* (%)	3515/4227 (45.4/54.6)	1455/1474 (49.7/50.3)	*p* < 0.001 (*χ*^2^ = 15.6)	620/607 (50.5/49.5)	*p* > 0.05 (*χ*^2^ =0.25)
Age (years), mean (SD)	56.7 (22.8)	57.1 (26.0)	*p* > 0.05 (*t* = −0.87)	53.1 (32.2)	*p* < 0.001 (*t* = −4.2)
Age of the deceased (years), mean (SD)	75.2 (12.2)	77.0 (13.0)	*p* = 0.03 (*t* = −2.2)	78.0 (13.3)	*p* > 0.05 (*t* = 0.76)
Number and percentage of patients in age groups (years), *n* (%)					
<20	837 (10.8)	376 (12.8)	*p* = 0.03 (*χ*^2^ = 8.7)	296 (24.1)	*p* < 0.001 (*χ*^2^ = 81.3)
20–40	723 (9.3)	316 (10.8)	*p* = 0.02 (*χ*^2^ = 5.1)	101 (8.2)	*p* = 0.01 (*χ*^2^ = 6.2)
40–60	2 127 (27.5)	607 (20.7)	*p* < 0.001 (*χ*^2^ = 50.8)	135 (11.0)	*p* < 0.001 (*χ*^2^ = 55.7)
60–80	3 090 (39.9)	1072 (36.6)	*p* = 0.002 (*χ*^2^ = 9.8)	405 (33.0)	*p* = 0.03 (*χ*^2^ = 4.9
>80	965 (12.5)	558 (19.1)	*p* < 0.001 (*χ*^2^ = 75.3)	290 (23.6)	*p* < 0.001 (*χ*^2^ = 11.2)
Presence of comorbidities *, *n* (%)					
Among all patients	5503 (71.1)	2038 (68.6)	*p* < 0.001 (*χ*^2^ = 40.6)	896 (73.0)	*p* = 0.03 (*χ*^2^ = 4.9)
Among the deceased	718/760 (94.5)	336/358 (93.9)	*p* > 0.05 (*χ*^2^ = 1.0)	113/118 (95.8)	*p* > 0.05 (*χ*^2^ = 0.9)

SD—standard deviation; *—most common comorbidities included hypertension, cardiovascular disease, type 2 diabetes, chronic obstructive pulmonary disease, and cancers.

**Table 2 viruses-15-00149-t002:** Characteristics of the course of the disease during periods with the dominance of particular SARS-CoV-2 variants.

Parameters	Pre-Delta 1 March 2020–30 June 2021 *N* = 7742	Delta 1 July 2021–31 December 2021 *N* = 2929	Delta vs. Pre-Delta	Omicron 1 January 2022–30 June 2022 *N* = 1227	Omicron vs. Delta
Need for oxygen therapy, *n* (%)	3849 (49.7)	1675 (57.2)	*p* < 0.001 (*χ*^2^ = 47.5)	451 (36.8)	*p* < 0.001 (*χ*^2^ = 144.5)
Need for mechanical ventilation, *n* (%)	441 (5.7)	197 (6.7)	*p* = 0.04 (*χ*^2^ = 4.0)	29 (2.4)	*p* < 0.001 (*χ*^2^ = 32.0)
Baseline SpO_2_ * < 91% or ARDS **, *n* (%)	2494 (32.2)	1131 (38.6)	*p* < 0.001 (*χ*^2^ = 38.8)	302 (24.6)	*p* < 0.001 (*χ*^2^ = 75.0)
Length of hospitalization, mean (SD)	11.7 (8.5)	11.2 (7.9)	*p* = 0.008 (*t* = 2.7)	9.3 (8.7)	*p* < 0.001 (*t* = 6.8)
Clinical status at the admission to the hospital, *n* (%)					
Asymptomatic	400 (5.2)	53 (1.8)	*p* < 0.001 (*χ*^2^ = 143.1)	54 (4.4)	*p* < 0.001 (*χ*^2^ = 23.2)
SpO_2_ ^1^ > 95%	2343 (30.3)	724 (24.7)	*p* < 0.001 (*χ*^2^ = 31.9)	552 (45.0)	*p* < 0.001 (*χ*^2^ = 167.0)
SpO_2_ ^1^ 91–95%	2342 (30.3)	946 (32.3)	*p* = 0.04 (*χ*^2^ = 3.9)	300 (24.5)	*p* < 0.001 (*χ*^2^ = 25.3)
SpO_2_ ^1^ < 91%	2429 (31.4)	1073 (36.7)	*p* < 0.001 (*χ*^2^ = 26.7)	292 (23.8)	*p* < 0.001 (*χ*^2^ = 65.6)
ARDS ^2^	65 (0.8)	57 (1.9)	*p* < 0.001 (*χ*^2^ = 23.0)	10 (0.8)	*p* = 0.008 (*χ*^2^ = 7.0)
Unknown	163 (2.1)	76 (2.6)	*p* > 0.05 (*χ*^2^ = 2.3)	19 (1.5)	*p* = 0.04 (*χ*^2^ = 4.2)
The most common symptoms, *n* (%)					
Cough	4876 (63.0)	2087 (71.3)	*p* < 0.001 (*χ*^2^ = 64.1)	582 (47.4)	*p* < 0.001 (*χ*^2^ = 213.5)
Fever	5441 (70.3)	2026 (69.2)	*p* > 0.05 (*χ*^2^ = 1.2)	677 (55.2)	*p* < 0.001 (*χ*^2^ = 74.5)
Dyspnoea	3803 (49.1)	1568 (53.5)	*p* < 0.001 (*χ*^2^ = 16.5)	356 (29.0)	*p* < 0.001 (*χ*^2^ = 209.1)
Loss of smell and taste	952 (12.3)	214 (7.3)	*p* < 0.001 (*χ*^2^ = 54.4)	28 (2.3)	*p* < 0.001 (*χ*^2^ = 39.8)
Diarhoea	898 (11.6)	400 (13.7)	*p* = 0.004 (*χ*^2^ = 8.4)	159 (13.0)	*p* > 0.05 (*χ*^2^ = 0.3)
Headache	1012 (13.1)	347 (11.8)	*p* > 0.05 (*χ*^2^ = 2.9)	93 (7.6)	*p* < 0.001 (*χ*^2^ = 16.6)
Nausea	500 (6.5)	217 (7.4)	*p* > 0.05 (*χ*^2^ = 3.1)	80 (6.5)	*p* > 0.05 (*χ*^2^ = 1.0)
Vomiting	410 (5.3)	218 (7.4)	*p* < 0.001 (*χ*^2^ = 17.8)	150 (12.2)	*p* < 0.001 (*χ*^2^ = 24.5)
Fatigue	2880 (37.2)	1441 (49.2)	*p* < 0.001 (*χ*^2^ = 127.0)	330 (26.9)	*p* < 0.001 (*χ*^2^ = 175.0)

^1^ SpO_2_—oxygen saturation; ^2^ ARDS—acute respiratory distress syndrome.

**Table 3 viruses-15-00149-t003:** Comparison of the frequency of discharge from hospitals during the 7, 14, 21, and 28 days of observation in the periods of dominance of particular variants.

	Pre-Delta 1 March 2020–30 June 2021 *N* = 7371	Delta 1 July 2021–31 December 2021 *N* = 2699	Delta vs. Pre-Delta	Omicron 1 January 2022–30 June 2022 *N* = 1220	Omicron vs. Delta
7 days, *n* (%)	1349 (18.3)	526 (19.5)	*p* > 0.05 (*χ*^2^ = 1.8)	446 (36.5)	*p* < 0.001 (*χ*^2^ = 131.2)
14 days, *n* (%)	4255 (57.7)	1361 (50.5)	*p* < 0.001 (*χ*^2^ = 42.7)	878 (71.9)	*p* < 0.001 (*χ*^2^ = 159.2)
21 days, *n* (%)	5487 (74.4)	1747 (64.8)	*p* < 0.001 (*χ*^2^ = 92.1)	1000 (82.0)	*p* < 0.001 (*χ*^2^ = 119.1)
28 days, *n* (%)	6079 (82.5)	2083 (77.2)	*p* < 0.001 (*χ*^2^ = 36.1)	1041 (85.3)	*p* < 0.001 (*χ*^2^ = 34.5)

**Table 4 viruses-15-00149-t004:** The 28-day mortality rates during periods with the dominance of particular SARS-CoV-2 variants.

Parameters	Pre-Delta 1 March 2020–30 June 2021	Delta 1 July 2021–31 December 2021	Delta vs. Pre-Delta	Omicron 1 January 2022–30 June 2022	Omicron vs. Delta
Overall 28-day mortality ^1^, *n* (%)	761/7371 (10.3)	359/2699 (13.3)	*p* < 0.001 (*χ*^2^ = 14.0)	119/1220 (9.8)	*p* = 0.005 (*χ*^2^ = 7.8)
Age-related mortality (years), *n*/*N* (%)					
<60	69/3542 (1.9)	37/1260 (2.9)	*p* = 0.04 (*χ*^2^ = 4.0)	9/512 (1.7)	*p* > 0.05 (*χ*^2^ = 1.9)
60–80	426/3118 (13.7)	199/1062 (18.7)	*p* < 0.001 (*χ*^2^ = 11.6)	49/390 (12.6)	*p* = 0.02 (*χ*^2^ = 5.6)
>80	340/1082 (31.4)	254/607 (41.8)	*p* < 0.001 (*χ*^2^ = 8.7)	74/316 (23.4)	*p* < 0.001 (*χ*^2^ = 15.4)
Mortality in subpopulations most at risk of death, *n*/*N* (%)					
SpO_2_ < 91%	561/2429 (23.1)	333/1074 (31.0)	*p* < 0.001 (*χ*^2^ = 14.2)	74/292 (25.3)	*p* > 0.05 (*χ*^2^ = 2.0)
Age >80 years +SpO_2_ < 91% +hypertension	164/359 (45.7)	131/221 (59.3)	*p* > 0.05 (*χ*^2^ = 3.2)	26/82 (31.7)	*p* = 0.01 (*χ*^2^ = 6.4)
Age > 80 years +SpO_2_ < 91% +COPD ^2^	16/40 (40.0)	17/31 (54.8)	*p* > 0.05 (*χ*^2^ = 0.6)	7/17 (41.2)	*p* > 0.05 (*χ*^2^ = 0.3)
Age > 80 years +SpO_2_ < 91% +diabetes	69/150 (46.0)	70/96 (72.9)	*p* = 0.03 (*χ*^2^ = 4.7)	19/40 (47.5)	*p* > 0.05 (*χ*^2^ = 1.8)
Age > 80 years +SpO_2_ < 91% +neoplasm	24/41 (58.5)	13/21 (61.9)	*p* > 0.05 (*χ*^2^ = 0.02)	9/14 (64.3)	*p* > 0.05 (*χ*^2^ = 0.01)
Age > 80 years +SpO_2_ < 91% +CIHD ^3^	81/153 (52.9)	53/82 (64.6)	*p* > 0.05 (*χ*^2^ = 0.8)	13/31 (41.9)	*p* > 0.05 (*χ*^2^ = 1.3)

^1^ Only data confirmed for 28 days of observation from the start of hospitalization were included; ^2^ chronic obstructive pulmonary disease; ^3^ chronic ischemic heart disease.

**Table 5 viruses-15-00149-t005:** Comparison of patients and disease characteristics during periods dominated by the BA1 and BA2 Omicron subvariants.

	BA1 *N* = 812	BA2 *N* = 408	*p*
Age (years), mean (SD)	50.0 (32.0)	59.1 (31.5)	*p* < 0.001 (*t* = 4.8)
SpO_2_ < 91% or ARDS, *n* (%)	194 (23.9)	106 (26.0)	*p* > 0.05 (*χ*^2^ = 0.6)
comorbidities, *n* (%)	579 (71.3)	315 (77.2)	*p* = 0.03 (*χ*^2^ = 4.8)
cough, *n* (%)	382 (47.0)	196 (48.0)	*p* > 0.05 (*χ*^2^ = 0.1)
fever, *n* (%)	452 (55.7)	224 (54.9)	*p* > 0.05 (*χ*^2^ = 0.1)
dyspnoea, *n* (%)	212 (26.1)	140 (34.3)	*p* = 0.003 (*χ*^2^ = 8.9)
need for oxygen therapy, *n* (%)	301 (37.1)	149 (36.5)	*p* > 0.05 (*χ*^2^ = 0.1)
need for mechanical ventilation, *n* (%)	20 (2,5)	9 (2,2)	*p* > 0.05 (*χ*^2^ = 0.1)
discharge within 7 days, *n* (%)	318 (39.2)	128 (31.4)	*p* = 0.008 (*χ*^2^ = 7.1)
discharge within 14 days, *n* (%)	582 (71.7)	296 (72.5)	*p* > 0.05 (*χ*^2^ = 0.1)
discharge within 21 days, *n* (%)	667 (82.1)	333 (81.6)	*p* > 0.05 (*χ*^2^ = 0.1)
discharge within 28 days, *n* (%)	697 (85.8)	344 (84.3)	*p* > 0.05 (*χ*^2^ = 0.5)
28-day mortality, *n* (%)	73 (9.0)	46 (11.3)	*p* > 0.05 (*χ*^2^ = 1.6)

## Data Availability

Data supporting reported results can be provided upon request from the corresponding author.

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
