# Peer review of "Variability in the Clinical Course of COVID-19 in a Retrospective Analysis of a Large Real-World Database"

_viruses, 2023, doi:10.3390/v15010149_

Round 1

Reviewer 1 Report

The authors report results from a real-life cohort of COVID-19 patients treated in 44 centers in Poland. 

The results of the study are interesting. However, I have major concerns regarding the experimental design of the study. The authors analysed data in three periods corresponding to the periods of dominants of various variants (pre-delta, delta, omicron) based on the information (not described specifically in the manuscript) on GISAID. However, no SARS-CoV-2 genotyping information was provided for the study participants. In my opinion, the concept of a study that is based on showing differences in clinically-relevant parameters of COVID-19 patients related to the variant wave but does not, at the same time, provide SARS-CoV-2 genotyping information on individual patients is not acceptable for publication in this form. Importantly, no information was provided on the pandemic waves in Poland in relation to the GISAID information. The process of variant replacement in different countries was probably not of identical kinetics and it is unclear how it can be matched to the GISAID data. The data on SARS-CoV-2 sequences from Poland indicates that sequencing information is available for the large group of COVID-19 patients with delta and omicron variants and it should provide an excellent data set for the analysis of the selected parameters in a cohort of patients with verified sequencing information. This approach could allow the authors to perform the analysis in a methodologically appropriate manner.

Reviewer 2 Report

  • Major comments: 

In the article, Robert Flisiak et al. provided a retrospective analysis of a nationwide Polish hospitalized patients database regarding the variability in the clinical course of COVID-19. They found that the currently dominant Omicron SARS-CoV-2 variant is more and more similar to the common cold disease with low pathogenicity.

The study is well-analyzed, and the authors also acknowledged the weak points of the study, including but not limited to the relatively arbitrary delineation of the pandemic period, the variant information of the patients was unknown, the ignorance of immunization or previous infection status, and, as the hospitalized patients were usually more severe cases, and the conclusion of the study could not generalize to the general population.

The study can be further improved by the following suggestions mentioned in specific comments.

  • Specific comments:

1)      In the 3. Results section, it would be better to use subtitles for each part of the conclusions to make the paper more organized.

2)      Line 122-123, grammar problem, consider changing to “Patient characteristics like sex, age and comorbidities were included in each analyzed period.”

3)      Line 165-166, it seems that the study cannot draw the conclusion that “The age of deceased patients was also significantly higher during the Delta compared to the pre-Delta period.” from table 1.

4)      In table 1, what comorbidities were included here?

5)      Line 176-177, it seems like that fever was not the most prevalent during the Delta period, according to table 2.

6)      Line 213-215, it would be better to include this data as a supplementary file.

7)      Line 222-225, it would be better to rephrase here, as it is misleading.

8)      Line 237, it should be age 20-80 here rather than “up to age 80,” according to table 1.

9)      Line 271-272, it should be noted here that the decreasing effects on mortality in diabetes and cardiovascular diseases were from different comparisons according to table 4.

Reviewer 3 Report

In this work, the authors describe an observational retrospective analysis of COVID-19 disease characteristics during three time periods roughly corresponding to the dominance of three different SARS-CoV-2 strains. The data indicate that in last time period (roughly corresponding to Omicron dominance) the disease was milder my multiple measures (WHO severity, hospitalization, symptoms). The authors suggest that this is due to an intrinsic weakening of the virus as the most plausible mechanism.

While this conjecture may indeed by correct, the authors need to account for multiple confounders in this observational analysis. In particular, note of the vaccination status of the individuals during each of the three time periods is of particular importance. If in unvaccinated (and uninfected) individuals the disease characteristics do become milder over time, then an intrinsic weakening of the virus is more plausible. Without these data on vaccination and prior infection it is difficult to determine whether the observed decrease in disease severity is due to host or to viral characteristics.

Nonetheless, as an observational study of COVID-19 disease characteristics during three time periods of the pandemic in Poland, the data are strong, subject to the limitations noted by the authors, even if the causes of the milder disease characteristics in the latter part of the pandemic remain obscure.

Specific recommendations:

(1) The introduction section can be reduced dramatically to perhaps two paragraphs with appropriate citations of prior literature on possible viral attenuation over time and the reasons for undertaking the study. A basic introduction to the disease is not needed.

(2) In the methods section there is commentary on why vaccination status was not analyzed. This belongs in the discussion.

(3) The discussion should at first center around the data and its limitations. After this, hypotheses about why the disease appears milder in the later timepoints should be entertained, and the data and outside literature should be discussed in support or against the various hypotheses.

(4) Finally, I am not sure it is completely appropriate to discuss each time point as "pre-Delta," "Delta," and "Omicron" without having actual viral sequence data.

Round 2

Reviewer 1 Report

The authors responded to all comments successfuly and invested a lot of time and effort in improving the manuscript. It is now suitable for publication. 

Reviewer 3 Report

My concerns have been addressed.